# Recent Progress in Iron-Based Microwave Absorbing Composites: A Review and Prospective

**DOI:** 10.3390/molecules27134117

**Published:** 2022-06-27

**Authors:** Wei Zheng, Wenxian Ye, Pingan Yang, Dashuang Wang, Yuting Xiong, Zhiyong Liu, Jindong Qi, Yuxin Zhang

**Affiliations:** 1China Academy of Space Technology (Xi’an), Institute of Space Antenna, Xi’an 710100, China; zhidaoyuan@163.com; 2School of Automation, Chongqing University of Posts and Telecommunications, Chongqing 400065, China; yewenxian123@163.com (W.Y.); xiongyt998@163.com (Y.X.); 15823521656@163.com (Z.L.); express0523@163.com (J.Q.); 3College of Material Science and Engineering, Chongqing University, Chongqing 400044, China; 20210901021@cqu.edu.cn

**Keywords:** Fe magnetic composites, microwave absorbing particles, controllable synthesis, microwave absorbing properties

## Abstract

With the rapid development of communication technology in civil and military fields, the problem of electromagnetic radiation pollution caused by the electromagnetic wave becomes particularly prominent and brings great harm. It is urgent to explore efficient electromagnetic wave absorption materials to solve the problem of electromagnetic radiation pollution. Therefore, various absorbing materials have developed rapidly. Among them, iron (Fe) magnetic absorbent particle material with superior magnetic properties, high Snoek’s cut-off frequency, saturation magnetization and Curie temperature, which shows excellent electromagnetic wave loss ability, are kinds of promising absorbing material. However, ferromagnetic particles have the disadvantages of poor impedance matching, easy oxidation, high density, and strong skin effect. In general, the two strategies of morphological structure design and multi-component material composite are utilized to improve the microwave absorption performance of Fe-based magnetic absorbent. Therefore, Fe-based microwave absorbing materials have been widely studied in microwave absorption. In this review, through the summary of the reports on Fe-based electromagnetic absorbing materials in recent years, the research progress of Fe-based absorbing materials is reviewed, and the preparation methods, absorbing properties and absorbing mechanisms of iron-based absorbing materials are discussed in detail from the aspects of different morphologies of Fe and Fe-based composite absorbers. Meanwhile, the future development direction of Fe-based absorbing materials is also prospected, providing a reference for the research and development of efficient electromagnetic wave absorbing materials with strong absorption performance, frequency bandwidth, light weight and thin thickness.

## 1. Introduction

The rapid development of communication technology in civil and military fields leads to various electromagnetic waves flooding peoples’ living environments, causing serious electromagnetic radiation pollution and bringing great harm [1,2,3]. To solve the electromagnetic radiation pollution caused by electronic equipment, one of its core technologies is the development of relevant electromagnetic wave absorbing materials (EMWAMs). For this reason, scholars have devoted themselves to the research of efficient EMWAMs in the last ten to twenty years [4,5].

The ideal EMWAMs should have the characteristics of “strong, wide, light and thin” (strong absorption of EM wave, wide effective absorption band, light weight and thin thickness) [6]. Currently, EMWAMs are divided into three categories according to the loss mechanism: dielectric, resistive and magneto-dielectric. Dielectric-type absorbing materials such as semiconductors and oxides own the advantages of low dielectric constants, good impedance matching properties, and chemical stability, but their electromagnetic loss capability is insufficient [7,8]. Resistive absorbing materials such as carbon materials have the advantages of high conductivity loss, low density and abundant dipoles, but they have poor impedance matching characteristics and high reflectivity; thus, it is difficult for electromagnetic waves to incident into the material, and a large number of electromagnetic waves are reflected into space, resulting in secondary pollution [9,10]. Magnetic metal wave-absorbing materials have excellent properties, such as high dielectric constant and permeability, high magnetic loss, high cut-off frequency, high saturation magnetization strength, and good temperature stability, which are a class of wave-absorbing materials with great development potential and application prospects [11,12,13]. Among them, Fe magnetic metal has the best performance, the most research, and the most extensive application. Fe magnetic absorbent particle material has good magnetic properties, high Snoek’s cutoff frequency, saturated magnetization, and Curie temperature, demonstrating an excellent electromagnetic wave loss ability, which has been an important direction for the development of microwave absorbing materials [14,15,16]. Although the conventional structure and single-component Fe magnetic absorbers have a certain wave absorption performance, Fe magnetic absorbent particles have poor impedance matching characteristics, high density, easy oxidation, and strong skin effect, resulting in the ideal EMWAMs’ performance that cannot be satisfied. In order to improve the comprehensive microwave absorption performance, the following two performance improvement strategies are generally used: one is to improve the magnetic permeability and magnetic loss of the material by a morphological structure design to enhance the wave absorption performance [17,18]; the other is to enhance the wave absorption performance by multi-component material compounding to enrich the loss mechanism and optimize the impedance matching [19,20]. Therefore, in recent years, the research of Fe metal magnetic absorbing materials is mainly focused on the preparation of different morphologies and other material composites.

In this paper, the research progress of Fe-based absorbing materials is reviewed firstly, and then the preparation methods, absorbing properties, and absorbing mechanism of Fe-based absorbing materials are discussed in detail from two aspects: different morphologies of Fe-based absorbing agents and Fe-based composite absorbing agents. Finally, the future development direction of Fe-based absorbing materials is prospected.

## 2. Wave Absorption Mechanism of EMWAMs

The absorbing mechanism of EMWAMs is shown in Figure 1. Due to the mismatch between the impedance of the free space and the impedance of the medium, when the electromagnetic wave propagates in space and meets the medium, part of the electromagnetic wave will be reflected at the interface between the free space and the medium, while the other part will be refracted into the medium. The electromagnetic waves propagating inside the medium will interact with the medium and convert the energy of electromagnetic waves into other forms of energy such as heat, electricity, mechanical energy, etc., for dissipation [21].

The impedance matching characteristics of the absorbing material can be achieved by creating and designing special boundary conditions. For a single-layer absorbing material model, when an electromagnetic wave is irradiated vertically from a free space with impedance $Z_{0}$ to an absorbing layer with input impedance $Z_{in}$, the reflection coefficient of the electromagnetic wave is:(1)$$R=\frac{Z_{0}-Z_{in}}{Z_{0}+Z_{in}}$$
(2)$$Z_{in}=\frac{E}{H}=\sqrt{\frac{\mu_{r}\mu_{0}}{\varepsilon_{r}\varepsilon_{0}}}$$
(3)$$Z_{0}=\frac{\mu_{0}}{\varepsilon_{0}}$$
where E and H denote the electromagnetic strength and magnetic field strength when there are electromagnetic waves in the material, respectively; $\mu_{0}$ and $\varepsilon_{0}$ are the free space permeability and dielectric constant, respectively; $\mu_{r}$ and $\varepsilon_{r}$ are the magnetic permeability and dielectric constant of the material, respectively [22].

When $Z_{in}\;$=$\;Z_{0}$, that is $\mu_{0}$/$\mu_{r}\;$=$\;\varepsilon_{0}$/$\varepsilon_{r}$, the reflection coefficient R = 0, the material achieves impedance matching with free space. In addition, for a specific wavelength of electromagnetic waves, the design thickness can be targeted d=nλ∕4 (*n* = 1, 3, 5, …). The absorbing layer (called the narrow-band resonant absorber layer) is where the phase difference between the reflected electromagnetic waves on both upper and lower surfaces of the absorbing layer is 180°; thus, that the interference cancellation makes R minimum [23].

The realization of the attenuation characteristics requires that the electromagnetic parameters of the absorbing material meet specified requirements. According to the microwave transmission line principle, the attenuation of electromagnetic waves per unit length of the material can be expressed in terms of the attenuation parameter α, which is expressed as follows [24]:(4)$$\mu_{r}=\mu^{\prime }-j\mu^{\prime \prime }$$
(5)$$\varepsilon_{r}=\varepsilon^{\prime }-j\varepsilon^{\prime \prime }\;$$
(6)$$\alpha =\frac{\sqrt{2}\pi f}{c}\times \sqrt{\left(\mu^{\prime \prime }\varepsilon^{\prime \prime }-\mu^{\prime }\varepsilon^{\prime }\right)+\sqrt{{\left(\mu^{\prime \prime }\varepsilon^{\prime \prime }-\mu^{\prime }\varepsilon^{\prime }\right)}^{2}+{\left(\mu^{\prime }\varepsilon^{\prime \prime }+\mu^{\prime }\varepsilon^{\prime \prime }\right)}^{2}}}\;$$

Equations (4) and (5) are the complex expressions of material permeability $\mu_{r}$ and dielectric constant $\varepsilon_{r}$, respectively. *f* and *c* are the frequency and propagation speed of the electromagnetic wave in a vacuum, respectively. According to Equation (6), it can be observed that to achieve attenuation of the incident electromagnetic waves, it is necessary to satisfy that $\mu^{\prime \prime }$ and $\varepsilon^{\prime \prime }$ are not simultaneously 0. Moreover, to achieve the efficient absorption of electromagnetic waves, it is necessary to increase the value of α. Therefore, $\mu^{\prime \prime }$ is always as large as possible and $\mu^{\prime }$ is as small as possible (there is no such problem in the electric loss absorbing materials), while $\varepsilon^{\prime \prime }$ and $\varepsilon^{\prime }$ are determined by the type of material; for electric loss absorbing materials, $\varepsilon^{\prime \prime }$ large and $\varepsilon^{\prime }$ small is as good; the magnetic loss absorbing materials are the opposite.

The permeability real part $\mu^{\prime }$ and dielectric constant real part $\varepsilon^{\prime }$ represent the storage capacity of magnetic field energy and electric field energy of the incident electromagnetic wave, respectively. The imaginary part of permeability $\mu^{\prime \prime }$ and the imaginary part of dielectric constant $\varepsilon^{\prime \prime }$ indicate the energy loss ability [25]. For the absorbing material, its absorbing performance is closely related to the electromagnetic parameters, and the loss factor is usually used to characterize the dielectric loss and magnetic loss of the material for electromagnetic waves, namely:(7)$$\tan\delta_{\varepsilon }=\varepsilon^{\prime \prime }/\varepsilon^{\prime }\;$$
(8)$$\tan\delta_{\mu }=\mu^{\prime \prime }/\mu^{\prime }\;$$

Equations (7) and (8) are the electric loss factor and magnetic loss factor of the material, respectively. It can be observed that the larger the $\varepsilon^{\prime \prime }$ and $\mu^{\prime \prime }$ of the wave-absorbing material, the stronger its ability to lose electromagnetic waves. In practical applications, all factors should be taken into consideration to improve the impedance and loss factor as well as the internal structure by selecting the material type (magnetic or dielectric) and thickness, so as to achieve the optimization of the absorber performance and obtain a high-performance absorber with thin thickness, light mass, wide frequency band, and complete functions.

The mechanism of electromagnetic attenuation caused by the interaction between absorbing materials and electromagnetic waves mainly includes the following three aspects: (1) High-frequency dielectric loss, electrical loss, hysteresis loss, or other forms of energy (thermal energy, electrical energy, mechanical energy, etc.) occur to make electromagnetic wave energy attenuation; (2) After the electromagnetic wave energy with a certain direction is affected by the absorbing material, it is transformed into electromagnetic energy dispersed in all possible directions, so that its intensity decreases sharply and the echo quantity decreases. (3) The first electromagnetic reflection wave acting on the material surface overlaps with the second electromagnetic reflection wave incident inside the material, making them interfere with each other and offset each other [26].

## 3. Preparation and Absorption Properties of Fe with Different Morphologies

The advantages of low cost, easy synthesis, high biodegradability, and biocompatibility of Fe make it a potential application material relative to other transition metals [27]. In the past decades of research, Fe has been widely used in various materials, such as wave absorbing materials [28], magnetic materials [29], catalyst materials [30,31], imaging materials [32] and detectors [33,34], etc.

It is well known that morphology, which includes microstructure, shape, and size, plays a crucial role in the properties of materials [35]. The particles with different morphologies have different specific surface areas, magnetic anisotropy fields, etc., which lead to the difference in interface effect, demagnetization field, and other parameters. When the particle size enters the nanometer scale, it brings about quantum effects that may lead to the splitting of electron energy that accompanies the formation of a new band gap, which also leads to the absorption of microwave energy; at the same time, nanomaterials with a high density of point defects (such as vacancies) and dangling bonds are prone to polarization in the electromagnetic wave field, which can consume some electromagnetic wave energy and contribute to enhanced wave absorption [36,37]. The variation of parameters caused by the morphology and size of the material has an important effect on its absorbing properties. Therefore, it is of great significance to summarize the preparation methods and absorbing properties of Fe with different morphologies for the preparation of new Fe absorbing materials with better comprehensive properties. Figure 2 summarizes the reported classification and preparation methods of different morphologies of Fe.

### 3.1. Sphere-Like Fe

At present, the commonly used methods for preparing spherical Fe are the one-step solvothermal method [35], corrosion method [38], chemical vapor condensation method [39], and gaseous nitridation method [40], etc. Scholars have prepared hollow and porous spherical Fe particle absorbents by various methods. Such a hollow and porous spherical Fe particle absorbent can not only reduce weight but also produce special morphology to improve electromagnetic properties [41]. Table 1 summarizes some commonly used preparation methods and absorption properties of spherical Fe absorbent reported.

Mingxu Sui et al., synthesized hollow Fe_3_O_4_ particles with a diameter of 200–1000 nm and shell thickness of 35–280 nm under different reaction conditions by the one-step solvothermal method (Figure 3a) for lightweight and efficient microwave absorption. The results demonstrate that the microwave absorption performance increases with the increase in hollow structure size, which is related to the reaction conditions. The sample prepared at 200 °C for 36 h had the best performance. The mass fraction of 70 wt% Fe_3_O_4_ was mixed with paraffin wax, as shown in Figure 3b. When the thickness was 2.07 mm, the frequency was 11.76 GHz, and the minimum reflection loss (RL_min_) was −55.14 dB. When the thickness is 2.07 mm, the effective absorption bandwidth reaches 4.72 GHz (5.6–10.32 GHz). The hollow structure makes Fe_3_O_4_ particles obtain a lower density and magnetic loss and optimized the impedance matching characteristics, so as to obtain an excellent microwave absorbing performance [35].

Guoxiu Tong et al., prepared porous iron particles (PIPs) through an easy corrosion technology (Figure 3c). The PIPs were made by corroding carbonyl iron powders (CIPs) with citric acid and ferric chloride, and the morphology of the PIPs was changed by controlling the concentration of the etching solution and the number of times of etching, (in Figure 3c, the 0.1 M citric acid and 0.2 M FeCl_3_ solution are used for etching for 15 min), and the complex permittivity, permeability and electromagnetic wave absorption properties of PIP were studied in the frequency range of 2 to 18 GHz. The results demonstrate that PIP with a content of 20 vol% is mixed with paraffin wax to make the sample. The matching thickness is in the range of 1.5–3 mm thickness, and the frequency is in the range of 7.2–17.2 GHz RL ≤ −20 dB. As shown in Figure 3d, when the thickness is 1.8 mm, RL_min_ at 13.2 GHz is −42.2 dB. The porous structure of PIPs increases the multi-polarization and multi-scattering of electromagnetic waves, thus enhancing the absorption performance, and has the potential for strong, wide, and light absorption materials [38]. Meijie Yu et al., prepared single-phase Fe_4_N particles at nanometer and micron levels by gaseous nitridation. Figure 3e shows the appearance of nanoscale Fe_4_N particles and compared the absorption performance of particles of different sizes in the range of 1–18 GHz. The results demonstrate that the dielectric constant of nanoparticles is higher than that of micron particles due to a large amount of interfacial polarization, and the absorption performance of the nanoparticle is higher than that of micron particles. The absorbing properties were tested by mixing Fe_4_N nanoparticles containing 75 wt% with paraffin wax to make composites with thicknesses ranging from 1.2 to 5.0 mm, with RL ≤ −10 dB in the frequency range of 1.8 to 11 GHz (as shown in Figure 3f). When the absorbent thickness is 3 mm, the RL_min_ value (RL_min_ = −33 dB) is obtained at 3.5 GHz [40].

Many spherical ferromagnetic absorbers generally have high permeability and magnetic loss and have good absorbing properties in the thickness range and frequency range of specific absorbers. Spherical ferromagnetic absorbers have the potential of excellent absorbing materials.

### 3.2. Flaky-Like Fe

At present, the ball milling method is generally used to prepare flake Fe particles. Compared with spherical particles, flake ferromagnetic particles can reduce eddy current loss (increase permeability) and increase space charge polarization (improve dielectric constant), thus improving the absorbing performance [42,43]. In the preparation of flake Fe, flake-shaped carbonyl iron (FCI) is the simplest and most widely used, and adopts ball milling technology [44]. Ball milling technology is a very effective method to improve the magnetic permeability of materials. This method can produce flake particles on the surface of carbonyl-iron particles (CIP) at a nano-scale, which can reduce the saturation magnetization value of CIP and improve the aspect ratio of flake particles, effectively improving microwave absorption properties [45,46]. Table 2 summarizes some reported absorption properties of FCI prepared from CIP by ball milling.

Hongyu Wei et al., prepared FCI (Figure 4a) by high energy ball milling at 200 °C for 2 h and mixed 70 vol% powders with paraffin to obtain the best permeability parameters. In Figure 4b, it can be observed that the real part reaches 3.20 at 2 GHz and the imaginary part reaches 1.61 at 6.2 GHz. The results show that the permeability and dielectric constant of the material can be modified by changing the morphology to optimize the microwave absorption performance [46].

Peicheng Ji et al., prepared scaly FCI with good impedance matching performance and absorption performance by grinding CIP with the rod milling method (Figure 4c) and studied the absorption performance of FCI prepared with different milling time in 1−18 GHz. The composite material was prepared by dispersing 85 wt% powders in paraffin matrix. In Figure 4d, when the thickness was 1.5 mm, the RL value less than −10 dB could be obtained at 4.5−8.5 GHz, and the RL_min_ value of −15.7 dB could be obtained at 6.0 GHz [47]. Cheng Guo et al., used a high-performance ball milling technology to produce commercial spherical carbonyl iron (PACI) particles into planar anisotropic carbonyl iron (PACI) particles. The PACI morphology is shown in Figure 4e. Under the external directional magnetic field, 70 wt% PACI was mixed into paraffin to make anisotropic composite materials. The lowest complex dielectric constant and the highest complex permeability could be obtained by orientation for 60 min. When the thickness was 3.25 mm, the RL value of −53.1 dB could be obtained at 2.09 GHz. RL ≤ −10 dB is in the frequency range of 1.54 to 2.93 GHz (see Figure 4f) [42]. Yonggang Xu et al., prepared lamellar carbonyl iron particles (FCI) by a two-step grinding process (morphology as shown in Figure 4g) and optimized the absorbing performance of the absorber at 8−18 GHz. When the absorbing agent filling amount is 50 wt% and the thickness is 1.47 mm, the RL value less than −10 dB can be obtained at 8−18 GHz (see Figure 4h) [48]. Dianliang Zheng et al., firstly prepared FCI by the grinding process, and then carried out a chemical corrosion process to optimize the shape of FCI. The dielectric constant and permeability of optimized FCIs (Figure 4i) increased slightly. When the absorbing agent filling amount was 40 wt% and the thickness was 0.8 mm, an RL value less than −8 dB can be obtained at 5.92–18 GHz (Figure 4j) [49]. Saichao Dang et al., prepared flake carbonyl iron particles (FCIPs) as absorbent by ball milling, as shown in Figure 4k. The absorption performance in the frequency range of 2−18 GHz and 26.5−40 GHz is studied. A three-layer planar ultra-wideband microwave absorber was designed. The RL value of the 6 mm thick absorber was less than −10 dB in 91% of the band (Figure 4l) [50].

### 3.3. Wire-Like Fe

One-dimensional wire-like Fe has the advantages of a small size and the large specific area, which can improve the anisotropy and resonance frequency of magnetic materials [51]. For example, Fe nanofibers, Fe nanowires, Fe nanochains, etc., all have good microwave absorption performance and are the focus of current research. Table 3 summarizes some reported common preparation methods and wave absorption properties of wire-like Fe absorbers.

At present, the metal salt high-temperature reduction method and chemical vapor condensation method are commonly used to prepare nanofibers, but these methods have the disadvantages of sensitive synthetic conditions and high cost [52]. Therefore, many scholars have started to investigate simpler, low-cost, and efficient methods for the preparation of Fe nanofibers. For example, Xiaogu Huang et al., prepared Fe_3_O_4_ nanofibers by the electrostatic spinning method (as shown in Figure 5a). The prepared Fe_3_O_4_ nanofibers have an anisotropy and excellent electromagnetic loss capacity within the frequency range of 2−18 GHz, which has the potential as a wave-absorbing material [53]. Qiangchun Liu et al., used pyrolysis to prepare isotropic Fe nanofibers with the morphology of Figure 5b. The absorbent with a thickness of 2 mm was prepared by dispersing 50 wt% Fe nanofibers in paraffin wax. The minimum RL value of −17.8 dB was obtained at 9.9 GHz, and the RL value of less than −10 dB was obtained at 7.3−11.7 GHz (Figure 5c) [54].

**Table 3 molecules-27-04117-t003:** The electromagnetic absorption performance of wire-like Fe, dendrite-like Fe, and cube shape-like Fe.

Samples	Methods	ƒ_E_ (GHz)	Thickness (mm)	Filling Ratio	RL_min_ (dB)	Reference
Fe_3_O_4_ nanofibers	Electrospinning	2−18	-	-	-	[53]
Fe nanofibers	Pyrolysis	9.9	2	50 wt%	−17.8	[54]
Fe nanowires	Situ reduction	1.3	3.5	50 vol%	−32	[55]
Fe NWs	Situ reduction	2.72	1.42	20 wt%	−44.67	[56]
Chain-like Fe NWs	Hydrothermal	3.68	3	20 wt%	−27.28	[57]
Dendrite-like α-Fe	Electric field-induced, Electrochemical reduction	10	1.9	70 wt%	−32.3	[58]
Dendrite-like α-Fe_2_O_3_	Hydrothermal	2.5	3	70 wt%	−25	[59]
Cube shape-like Fe	Low-temperature solution reduction	9.1	2	26 vol%	−56	[60]

In situ reduction is a common method for the synthesis of Fe nanowires, which has the advantages of simplicity, high efficiency, and low cost, and is suitable for large-scale production. Fe nanowires are mainly prepared by reducing iron salt with sodium borohydride (NaBH_4_). For example, Xinghua Li et al., synthesized spherical and linear Fe nanowires by the NaBH_4_ reduction method (linear morphology is shown in Figure 6a). Compared with iron nanospheres, the magnetic conductivity, dielectric constant, and microwave absorption performance of iron nanowires are significantly improved. In Figure 6b, the minimum RL value of −32 dB was obtained at 1.3 GHz, and the RL value of less than −10 dB could be obtained at 0.8−2.1 GHz [55]. Ping-an Yang et al., synthesized iron nanowires (Fe NWs) with high-aspect-ratio, uniform length of about 21 μm, and a diameter of about 60 nm, which were synthesized by a magnetic field-induced in situ reduction method, and their electromagnetic properties in the frequency range of 2−18 GHz were investigated [48]. In addition to the in situ reduction method for the preparation of Fe NWs, the hydrothermal method is also used for the synthesis of Fe NWs. For example, Junyao Shen et al., synthesized necklace-like Fe NWs with a high aspect ratio of 100 nm in the average diameter by a magnetic field-assisted hydrothermal method (morphology is shown in Figure 6e) and tested the absorption performance of the sample with its content of 20 wt% at 2−18 GHz. The results demonstrated that Fe NWs have excellent absorbing performance in the range of 2−6 GHz. When the thickness is 3 mm, the RL value of 3.68 GHz in Figure 6f reaches −27.28 dB, which provides a reference for the study of low-frequency absorbing materials [56].

### 3.4. Dendrite-Like Fe and Cube Shape-Like Fe

At present, in addition to the common spherical, flaky, and linear forms, other complex morphologies of Fe absorbers have been prepared, such as dendritic and cubic forms, and Table 3 summarizes some of the commonly reported methods for preparing dendritic and cubic Fe absorbers and their wave absorption properties. For example, Zhenxing Yu synthesized three-dimensional dendritic α-Fe with a width of about 3.0 mm and a length of about 9.0 mm by electric field induction and electrochemical reduction, and the morphology is shown in Figure 6g. A total of 70 wt% samples were dispersed in paraffin to prepare a thickness of 1.9 mm absorber, which can achieve the minimum RL value of −32.3 dB at 10 GHz. When the absorbing thickness is 1.5 mm, the absorbing bandwidth (RL ≤ −10 dB) is 12 GHz (Figure 6h) in the frequency range of 6−14 GHz [58]. Genban Sun et al., firstly prepared the dendritic α-Fe_2_O_3_ by the hydrothermal method, and then obtained dendritic Fe particles by hydrogen reduction at high temperature, as shown in Figure 6i. The sample with a particle content of 70 wt% and thickness of 3 mm was found to have an RL value of −25 dB at 2.5 GHz by the absorption test (Figure 6j) [59]. Xi’an Fan et al., prepared cuboidal single-crystal Fe particles by the low-temperature solution reduction method, which demonstrated an excellent absorbing performance (morphology and RL plot as in Figure 6k,l). When the content is 26 vol% and the thickness is 2 mm, the RL value of −56 dB can be obtained at 9.1 GHz [60].

## 4. Preparation and Wave Absorption Properties of Fe Matrix Composites

Because of the strong sensitivity of pure iron, the air condition has a great influence on it and the impedance matching characteristic is poor. In order to develop high-performance absorbing materials, Fe-based composite materials have aroused many scholars to explore. Therefore, in recent years, the research of iron matrix composites has increased greatly. So far, Fe matrix composites have been widely studied and are ideal for numerous applications. Composite Fe particles with other materials is an effective way to suppress the eddy current effect, enhance absorbing efficiency, expand absorbing bandwidth and reduce the weight of the absorbing layer [61]. Currently, common methods for preparing Fe matrix composites include blending and surface coating (as shown in Figure 7).

### 4.1. Blending

Directly mixing pure Fe particles with other wave-absorbing materials is the simplest way to prepare Fe-based composites. Pure Fe particles can obtain more heterogeneous interfaces by coupling different materials, and heterogeneous interfaces provide more interface losses, which are beneficial to improve the electromagnetic absorbing ability. The mechanism is mainly the synergy of impedance matching and polarization loss.

Table 4 summarizes some reported composites prepared by mixing pure Fe with additional materials and their wave absorption properties. For example, Wongyu Jang et al., used the Doctor Blade method to disperse CIP with different mass fractions in the matrix of polydimethylsiloxane (PDMS) to prepare absorbent to study the absorbent performance within the frequency range of 0.1−18 GHz. When the CIP content is 72 wt% and the absorbent thickness is 1.5 mm, the absorption performance is the best, and the RL_min_ value of −27.5 dB can be obtained at 14.6 GHz. The morphology and RL plot are shown in Figure 8a,b [62]. Baoshun Zhu et al., used waste fly ash (FA) and Fe particles to carry out the carbothermal reduction process. Fe particles were evenly embedded into the interior and surface of the matrix to form composite materials with morphology, as shown in Figure 8c. The sample with a composite content of 65 wt% and a thickness of 2.5 mm was subjected to the wave absorption test. It was found that the RL value at 16.1 GHz reached −35.7 dB and the effective bandwidth reached 4.1 GHz. The curve of RL is shown in Figure 8d [63]. Kaichuang Zhang et al., used in situ polymerization to prepare MWCNTs/Fe_3_O_4/_PPY/C composites with 8.8% mass fraction of ferric oxide, carbon nanotubes, polypyrrole, and carbon. The morphology is shown in Figure 8e. The composite has multiple interfaces and multilayer structures, magnetic and dielectric losses, and excellent absorbing properties. The results demonstrate that the sample is prepared by mixing 25 wt% composite material with paraffin wax, when the matching thickness is 2.2 mm, the RL_min_ obtained at 13.92 GHz is −53.07 dB, and the effective absorption bandwidth is 6.4 GHz. The plot of RL is shown in Figure 8f [64].

At present, the preparation of biomaterial-based electromagnetic wave absorbing materials has become a research hotspot due to their advantages of low cost and no pollution, and renewable and easy processing [65,66]. For example, Xinfeng Zhou et al., synthesized the porous foam matrix with fish skin as raw material by the hydrothermal method, and then embedded Fe_3_O_4_ nanospheres into the carbon matrix uniformly through the reflux and annealing treatment, and obtained the new Fe_3_O_4_/C composite foam. The morphology and RL plot are shown in Figure 8g,h. The 25 wt% of the composite foam was dispersed in paraffin wax to make the absorbing specimen, and when the matched thickness was 1.9 mm, the RL_min_ value of −47.3 dB was obtained in the range of 12.2−17.8 GHz; when the matched thickness was 2.2 mm, the effective absorption bandwidth was 5.68 GHz (12.16−17.84 GHz) [65]. Figure 9 provides the electromagnetic wave absorption mechanism. The porous foam structure increases the multiple reflections of the incident electromagnetic wave, and the Fe_3_O_4_ and C mosaic mode increase the interfacial polarization and dipole polarization to increase the dielectric loss and magnetic loss, so as to attain efficient electromagnetic wave absorption.

### 4.2. Surface Coating

Core-shell structures of different sizes are formed through a surface coating, and such core-shell structures have rich heterogeneous interfaces, good conductive network, high anisotropy ratio, magnetic-dielectric synergy and other mechanisms [67,68,69,70]. The coating of pure Fe particles mainly includes the carbon coating [71], metal coating [72], semiconductor coating [73], and conductive polymer coating [74]. Table 5 summarizes some reported surface coating methods of Fe particles and Fe-based composites and their absorbing properties.

#### 4.2.1. Carbon Material-Coated Fe

Carbon material as a dielectric material has received much attention because of its small density, stable physicochemical properties, and strong conductivity loss [75]. Due to the easy oxidation and poor impedance matching of pure Fe particles, many scholars use carbon materials to encapsulate pure Fe particles to improve easy oxidation and enhance magnetic-dielectric synergy to optimize impedance matching characteristics. Various crystalline states of carbon materials, such as carbon nanofiber [71], graphene [76], and carbon nanotube [77] coated with pure Fe particles, can achieve excellent absorbing properties.

For example, Tao Wang et al., prepared Fe-C nanofiber composites with magnetic iron nanoparticles uniformly dispersed along the fibers and wrapped by carbon matrix using a electrostatic spinning technique, and the morphology and RL plot are shown in Figure 10a,b. The results demonstrated that when the Fe-C nanofibers with a mass fraction of 72 wt% were mixed with paraffin to prepare the sample, the RL_min_ value of −44 dB was obtained at 4.2 GHz when the matching thickness of the sample was 3 mm. When the thickness is 5.2−1.2 mm and the frequency range is 2.2−13.2 GHz, there is RL < −10 dB, indicating that Fe-C nanofibers’ composites have good absorption properties in the S-X band [71]. Seunggeun Jeon et al., realized the graphene oxide (GO) sheet covered with CIP (GO@CIPs) through the wet stirring process. The morphology and RL plot are shown in Figure 10c,d. The complex permeability and permittivity in the frequency range of 0.1−18.0 GHz were measured. The results demonstrate that by mixing 72 wt% of GO@CIPs powder with paraffin wax to make absorbers, RL_min_ values of −56.4 dB and −33.0 dB were obtained at 5.1 GHz and 4.8 GHz when the absorbers were 1.9 mm and 2 mm thick, respectively [78]. Xueai Li et al., successfully synthesized carbon-coated iron nanoparticles (Fe@C) with a rose-like porous structure (as in Figure 10e) by an in situ method using iron alcohol salt precursors as raw materials, and tested their wave absorption properties in the frequency range of 2.0 to 18.0 GHz. The results demonstrate that the wave absorbing specimens made by mixing Fe@C with paraffin wax in the ratio of 1:1 by weight, as in Figure 10f, achieved an RL_min_ value of −71.47 dB at 11.6 GHz when the specimens were matched with a thickness of 1.48 mm [79]. Figure 10g shows the electromagnetic wave propagation and attenuation process of Rose-like Fe@C. Firstly, the porous structure and the core-shell structure of Fe@C can be synergized to obtain a good impedance match so that more electromagnetic waves can enter, secondly, the porous structure and the three-dimensional network structure make the electromagnetic waves reflect and scatter many times inside, which is conducive to attenuating more electromagnetic waves. Then, the core-shell structure of Fe@C increases multiple interfacial polarization, realizing more conversion of the electromagnetic wave into heat energy, and the effect of the absorbing wave is significantly enhanced.

#### 4.2.2. Metal-Coated Fe

Metal Al produces a dense aluminum oxide (Al_2_O_3_) film when the oxidation reaction occurs, which can insulate the air to play an antioxidant role [80]. Some scholars have coated aluminum on the surface of oxidation-prone absorbing materials to achieve the absorbing and antioxidant-integrated absorbing materials. For example, Yingying Zhou et al., prepared nano aluminum (Al)-coated carbonyl iron particles (CIPs) by the ball milling method to prepare Al@CIPs absorbers with oxidation resistance, high absorption, and heat resistance. The results demonstrated that the absorbing specimens were made by mixing 70 wt% content of the powder with paraffin wax, and the RL_min_ value of −27.2 dB was obtained at 10.5 GHz when the thickness was 1.6 mm [81]. Metallic Ag has good electrical conductivity and stability, and Ag nanoparticles enhance interfacial polarization, dipole polarization, and conductivity loss, thus improving microwave absorption performance [82]. Ping-an Yang et al., synthesized Ag-coated Fe (Fe@Ag) core-shell nanowires with strong electromagnetic wave absorption, which were synthesized by a liquid-phase reduction and layer-plus-island growth methods (see Figure 11c), and the absorbing properties were investigated in the frequency range of 2.0 to 18.0 GHz. The results demonstrate that the comprehensive absorbing performance of the absorber is the best when the mass fraction of Fe@Ag (Fe:Ag ratio is 2:1) is mixed with paraffin wax. When the matching thickness is 3.36 mm, the RL_min_ value of −58.69 dB is obtained at 7.53 GHz. When the matching thickness is 2.93 mm, the effective absorption bandwidth is 7.32 GHz, covering the whole C and X bands [83]. Figure 11e provides the electromagnetic wave absorption mechanism of Fe@Ag core-shell NWs. First, the Ag shell increases the magnetic loss, adjusts the impedance matching, and can make more electromagnetic wave incident into the material; second, one-dimensional FeNWs with a high-aspect-ratio form a conductive network, resulting in the multiple reflection of incident electromagnetic waves, increasing the multiple reflection loss. Third, the Fe@Ag core-shell NWs structure forms multiple interfacial polarization, resulting in the conversion of electromagnetic waves into heat energy to consume more incident electromagnetic waves.

#### 4.2.3. Semiconductor-Coated Fe

Semiconductor materials such as ZnO [84], MnO_2_ [85], and SiO_2_ [86] have good dielectric properties and chemical stability. The semiconductor material coated on the surface of the magnetic metal Fe can improve the dielectric constant, optimize impedance matching and improve the absorbing performance [87,88,89,90].

ZnO, a dielectric material with excellent dielectric properties, is often combined with other materials for wave absorption to improve the performance of absorbing materials [84,91]. For example, Qi Liu et al., prepared iron/zinc oxide (Fe/ZnO) nanocomposites with a core-shell structure (see Figure 12a) using a low-temperature wet chemical method and investigated the wave-absorbing properties in the frequency range of 2.0 to 18.0 GHz. The results demonstrate that Fe/ZnO nanocomposites with a mass fraction of 50 wt% (Fe/Zn ratio of 1:0.75) are mixed with paraffin uniformly to make the sample, as shown in Figure 12b. When the matching thickness is 1.59 mm, the RL_min_ value of −48.28 dB is obtained at 15.55 GHz. When the matching thickness is 1.90 mm, the effective absorption bandwidth is 5.10 GHz (10.79−15.89 GHz) [92]. MnO_2_ is a kind of electrically dissipative material with good dielectric properties and chemical stability, which is commonly used as an absorbing material medium [93,94,95]. For example, Zhengwei Qu et al., prepared ultrathin MnO_2_ nanosheets-coated CIP spherical flower-like composites (CIP@MnO_2_) by a redox reaction (see Figure 12c), to improve the oxidation resistance and wave absorbing properties of CIP, and the absorbing properties of CIP@MnO_2_ with different filling ratios in the frequency range of 2.0−18.0 GHz were investigated. The results demonstrate that the absorption agent made by mixing 40 wt% CIP@MnO_2_ powder with paraffin wax has the best comprehensive performance. When the matching thickness is 10 mm, the RL_min_ at 6.32 GHz reaches −63.87 dB, and the effective absorption bandwidth (RL < −20 dB) reaches 7.28 GHz [96]. Figure 12e provides the wave absorption mechanism of CIP@MnO_2_ composites. The synergistic effect of multiple reflection, good conductive network, multiple interfaces polarization, magnetic coupling, and magnetic loss makes them have an excellent electromagnetic wave absorption performance.

#### 4.2.4. Conductive Polymer-Coated Fe

Conductive polymers have been extensively studied due to their high electrical conductivity [97,98]. When the conductive polymer exhibits the electrical conductivity in a wide range, the combination of the conductive polymer and the magnetic loss material produces a synergistic effect, which can not only make the composite material have special properties such as light weight, high temperature resistance and oxidation resistance, but also enrich the polarization loss and enhance the microwave absorption performance [99,100,101].

For example, Zhengchen Wu et al., prepared the polypyrrole (PPy) Fe_3_O_4_-coated core-shell Fe_3_O_4_@PPy composite material (Figure 13a) through corrosion, polymerization, replication and other processes, and evenly mixed 50 wt% Fe_3_O_4_@PPy powder with paraffin wax. In Figure 13b, when the matching thickness was 2.0 mm, the RL_min_ of −41.9 dB was obtained at 13.3 GHz, and the absorption bandwidth covered the entire Ku band. Figure 13c shows the possible electromagnetic wave absorption mechanism of the Fe_3_O_4_@PPy composite material [101]. Xiang Luo et al., prepared polyaniline-coated Fe_3_O_4_ dendritic Fe_3_O_4_@PANI composite by the hydrothermal method (Figure 13d). PANI can optimize impedance matching and form conductive networks to optimize the absorbing performance. When the matching thickness is 1.3 mm, the RL_min_ value of −53.08 dB is obtained at 3.04 GHz, and it has a wide EAB (4.1 GHz) [102]. Figure 13f provides the absorbing mechanism of dendritic Fe_3_O_4_@PANI composite material, which is mainly divided into three aspects: dielectric loss, magnetic loss, and good conductive network. The Fe_3_O_4_@PANI interfacial polarization occurs at heterogeneous interfaces and dipole polarization occurs at electric fields, which constitute dielectric loss. The natural resonance and exchange resonance of Fe_3_O_4_ itself constitutes a magnetic loss. The dendritic structure forms a good conductive network, which makes the electromagnetic wave reflect and scatter. The three ways work together to convert the incident electromagnetic wave into heat energy as much as possible in the composite material to consume the incident electromagnetic wave.

**Table 5 molecules-27-04117-t005:** The electromagnetic absorption performance of surface-coated Fe-based composites.

Samples	Methods	ƒ_E_ (GHz)	Thickness (mm)	Filling Ratio (wt%)	RL_min_ (dB)	Reference
Fe-C nanofibers	Electrospinning	4.2	3	-	−44	[71]
GO@CIPs	Wet stirring	5.1	1.9	72	−56.4	[78]
Fe@C	Situ reduction	11.6	1.48	50	−71.47	[79]
Al @CIPs	Ball milling	10.5	1.6	70	−27.2	[81]
Fe@Ag	Liquid-phase reduction etc.	7.53	3.36	25	−58.69	[83]
Fe/ZnO	Low-temperature wet chemical	15.55	1.59	50	−48.28	[92]
CIP@MnO_2_	Redox reaction	6.32	10	40	−63.87	[96]
Fe_3_O_4_@PPy	Corrosion, etc.	13.3	2.0	50	−41.9	[101]
Fe_3_O_4_@PANI	Hydrothermal	3.04	1.3	60	−53.08	[102]

## 5. Conclusions

On the basis of the present research, the preparation methods, absorption properties and absorption mechanism of representative Fe-based absorbing materials in recent years are summarized and discussed. Firstly, this paper briefly describes the electromagnetic wave absorption mechanism; secondly, the preparation methods and absorbing properties of the Fe wave absorbers with various morphologies were introduced from spherical, flake, linear and other morphologies. Then, the preparation and absorbing properties of the Fe-based composite absorbing materials are introduced by blending and surface coating. Finally, representative electromagnetic wave absorbers of each type are introduced, and the electromagnetic wave absorption mechanism of Fe-based absorbing materials is introduced.

As a magnetic metal, Fe has low cost, high saturation magnetization, Snoek’s limit, and Curie temperature, demonstrating excellent electromagnetic wave loss ability, so it is commonly used in the field of electromagnetic absorption. Only using the energy of conventional Fe magnetic absorbing materials cannot meet the need ideal absorbing materials. Therefore, various morphologies and structures of Fe magnetic particles have attracted people’s attention, and complex morphologies and nanostructures have been designed to enhance the performance, such as preparing pure Fe into porous and hollow spheres to increase reflection and scattering propagation paths, milling CIP spheres into flakes to reduce saturation magnetization values and improve aspect ratios, preparing pure Fe into nanowire-like structures with small size and large specific area, and other morphologies in dendritic and cubic shapes, which can exhibit excellent electromagnetic wave absorption properties. However, pure Fe of a single component will be limited by impedance matching and stability. Therefore, it will be a good solution to combine Fe magnetic particles with other materials to prepare composite materials.

The simplest way is to mix Fe magnetic particles with other materials to prepare composite materials, so as to combine the advantages of both to improve the absorption performance. Then, dielectric materials and metal materials such as carbon, a semiconductor, and conductive polymer are selected to cover the surface of pure Fe to form a core-shell structure. On the one hand, it can prevent the oxidation of pure Fe, on the other hand, it has rich heterogeneous interface, forms good conductive network, improves the anisotropy ratio, promotes magnetic-dielectric synergy and improves impedance matching to improve the comprehensive absorbing performance. In future research, multifunctional Fe-based absorbing materials will be the focus of research, such as thermal absorbing materials applied in a high temperature environment, anti-corrosion absorbing materials applied in an acid-base environment, absorbent flexible wearable devices used in medical care, and sensing fields. At the same time, the broadband absorption of electromagnetic waves will be the focus of research, and the broadband absorbing material is the key problem of the country, which is of great significance for the construction of the broadband stealth weapon platform.

## Figures and Tables

**Figure 1 molecules-27-04117-f001:**
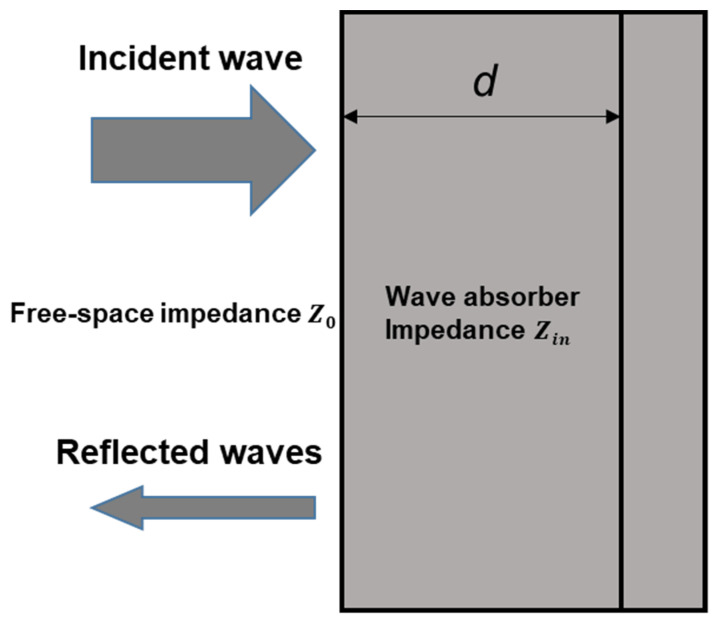
Schematic diagram of electromagnetic wave incident on the absorbent surface.

**Figure 2 molecules-27-04117-f002:**
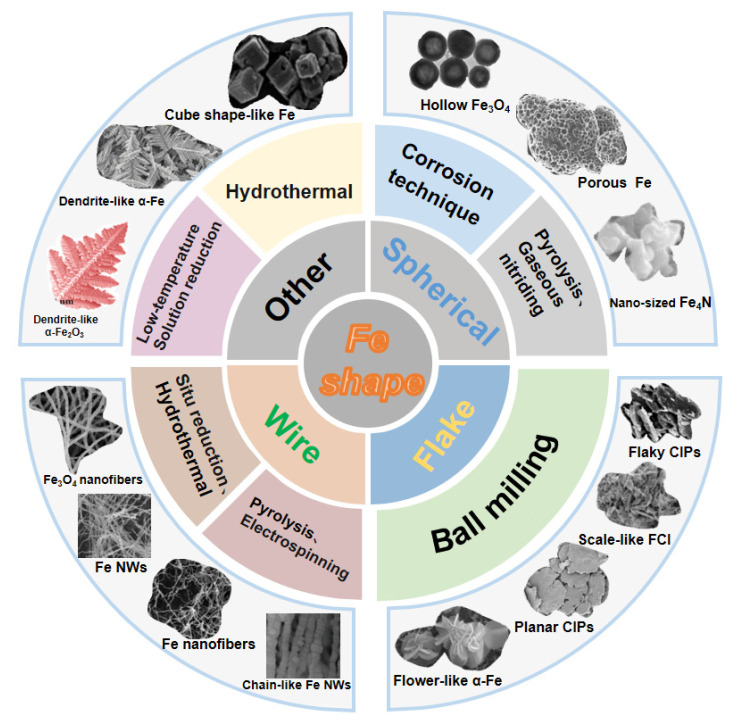
Classification and preparation of partially recent representatively different forms of Fe.

**Figure 3 molecules-27-04117-f003:**
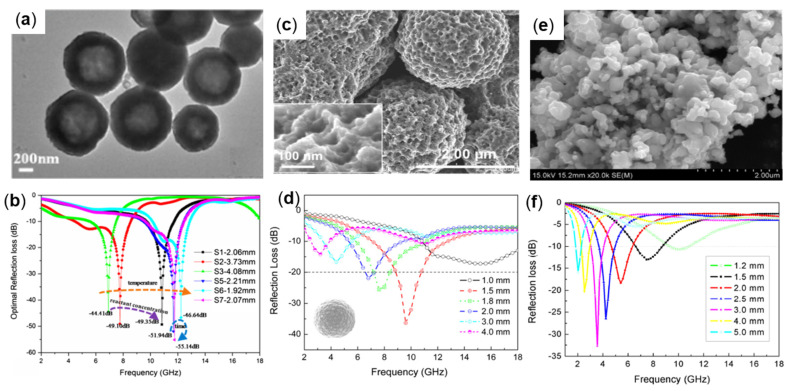
(**a**) SEM and (**b**) RL plot of hollow Fe_3_O_4_, reproduced with permission from Ref. [35], Copyright 2018, Springer Nature. (**c**) SEM and (**d**) RL plot of PIPs, reproduced with permission from Ref. [38], Copyright 2012, Elsevier. (**e**) SEM and (**f**) RL plot of hollow iron spiral particles, reproduced with permission from Ref. [40], Copyright 2016, Elsevier.

**Figure 4 molecules-27-04117-f004:**
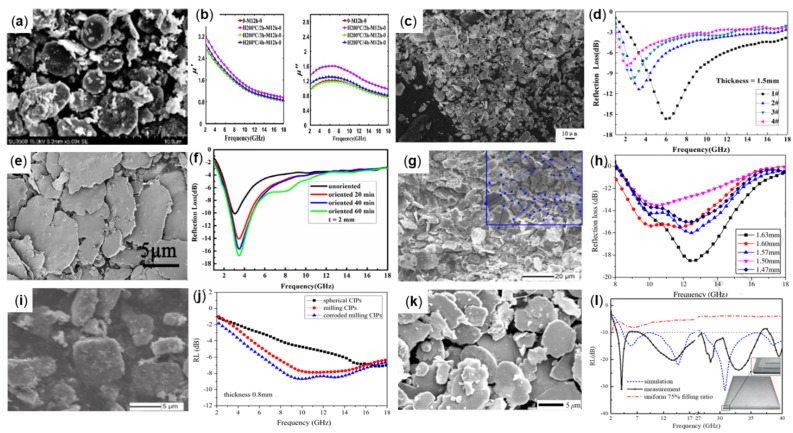
(**a**) SEM and (**b**) magnetic permeability plot of FCI, reproduced with permission from Ref. [46], Copyright 2020, Elsevier. (**c**) SEM and (**d**) RL plot of scale-like FCI, reproduced with permission from Ref. [47], Copyright 2019, Springer Nature. (**e**) SEM and (**f**) RL plot of PACI, reproduced with permission from Ref. [42], Copyright 2018, Elsevier. (**g**) SEM and (**h**) RL plot of FCI, reproduced with permission from Ref. [48], Copyright 2016, Elsevier. (**i**) SEM and (**j**) RL plot of FCI, reproduced with permission from Ref. [49], Copyright 2016, Elsevier. (**k**) SEM and (**l**) RL plot of FCIPs, reproduced with permission from Ref. [50], Copyright 2018, Springer Nature.

**Figure 5 molecules-27-04117-f005:**
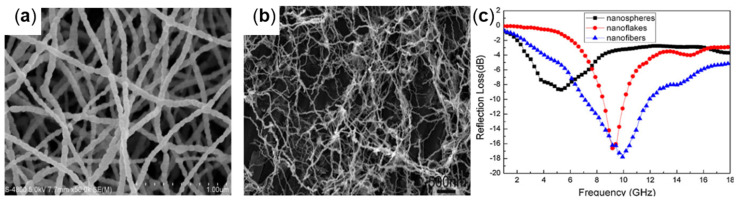
(**a**) SEM of Fe_3_O_4_ nanofibers, reproduced with permission from Ref. [53], Copyright 2015, Springer Nature. (**b**) SEM and (**c**) RL plot of Fe nanofibers, reproduced with permission from Ref. [54], Copyright 2011, Springer Nature.

**Figure 6 molecules-27-04117-f006:**
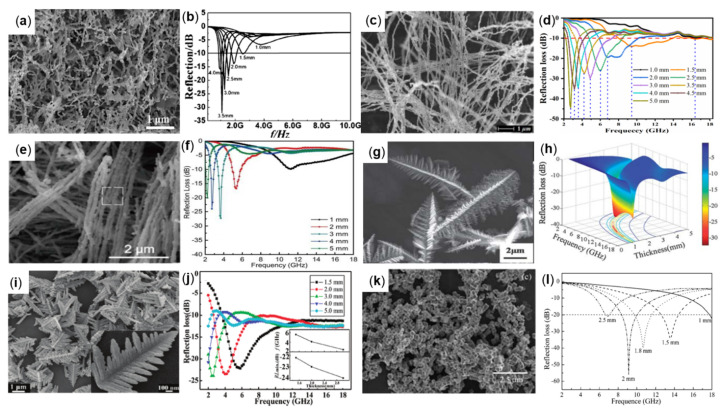
(**a**) SEM and (**b**) RL plot of Fe nanowires, reproduced with permission from Ref. [55], Copyright 2014, Elsevier. (**c**) SEM and (**d**) RL plot of Fe NWs, reproduced with permission from Ref. [56], Copyright 2020, IOP Publishing. (**e**) SEM and (**f**) RL plot of chain-like Fe NWs, reproduced with permission from Ref. [57], Copyright 2016, The Royal Society of Chemistry. (**g**) SEM and (**h**) RL plot of chain-like Fe NWs, reproduced with permission from Ref. [58], Copyright 2013, The Royal Society of Chemistry. (**i**) SEM and (**j**) RL plot of dendrite-like α-Fe_2_O_3_, reproduced with permission from Ref. [59], Copyright 2011, American Chemical Society. (**k**) SEM and (**l**) RL plot of cube shape-like Fe, reproduced with permission from Ref. [60], Copyright 2010, The Royal Society of Chemistry.

**Figure 7 molecules-27-04117-f007:**
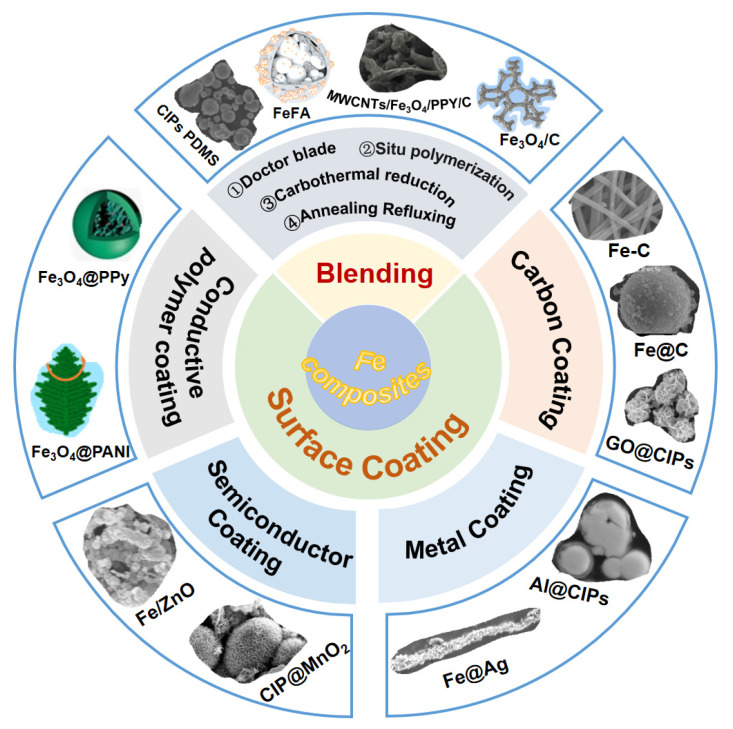
Classification and preparation of partially recent representative Fe-based composites.

**Figure 8 molecules-27-04117-f008:**
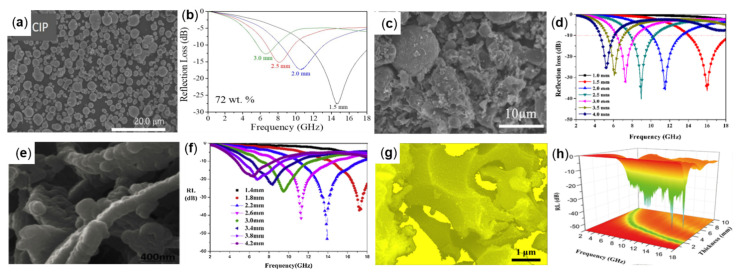
(**a**) SEM and (**b**) RL plot of CIPsPDMS, reproduced with permission from Ref. [62], Copyright 2020, New Physics Sae Mulli. (**c**) SEM and (**d**) RL plot of FeFA, reproduced with permission from Ref. [63], Copyright 2020, American Chemical Society. (**e**) SEM and (**f**) RL plot of MWCNTs/Fe_3_O_4_/PPY/C, reproduced with permission from Ref. [64], Copyright 2020, Elsevier. (**g**) SEM and (**h**) RL plot of Fe_3_O_4_/C, reprinted with permission from Ref. [65], Copyright 2019, Elsevier.

**Figure 9 molecules-27-04117-f009:**
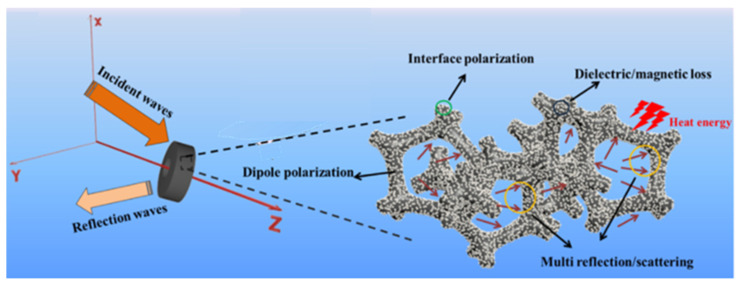
Schematic illustration for the electromagnetic wave absorption mechanism of Fe_3_O_4_/C samples. Reproduced with permission from Ref. [65], Copyright 2019, Elsevier.

**Figure 10 molecules-27-04117-f010:**
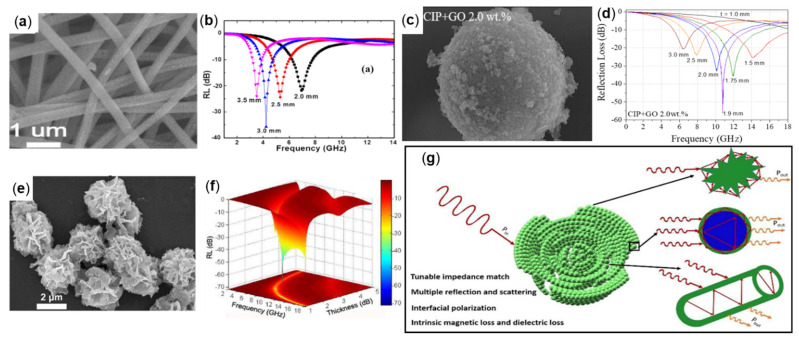
(**a**) SEM and (**b**) RL plot of Fe-C, reproduced with permission from Ref. [71], Copyright 2014, Elsevier. (**c**) SEM and (**d**) RL plot of GO@CIPs, reproduced with permission from Ref. [78], Copyright 2019, Elsevier. (**e**) SEM and (**f**) RL plot of Fe@C and (**g**) schematic illustration for the electromagnetic wave absorption mechanism of rose-like Fe@C, reproduced with permission from Ref. [79], Copyright 2018, The Royal Society of Chemistry.

**Figure 11 molecules-27-04117-f011:**
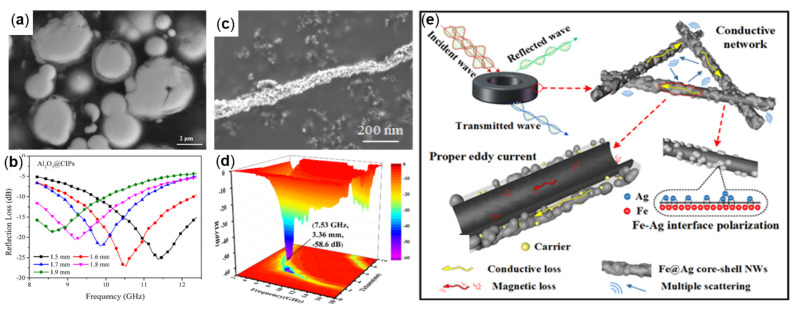
(**a**) SEM and (**b**) RL plot of Al @CIPs, reproduced with permission from Ref. [81], Copyright 2021, Elsevier. (**c**) SEM and (**d**) RL plot of Fe@Ag and (**e**) EM wave attenuation mechanism of Fe@Ag core-shell NWs, reproduced with permission from Ref. [83], Copyright 2022, Elsevier.

**Figure 12 molecules-27-04117-f012:**
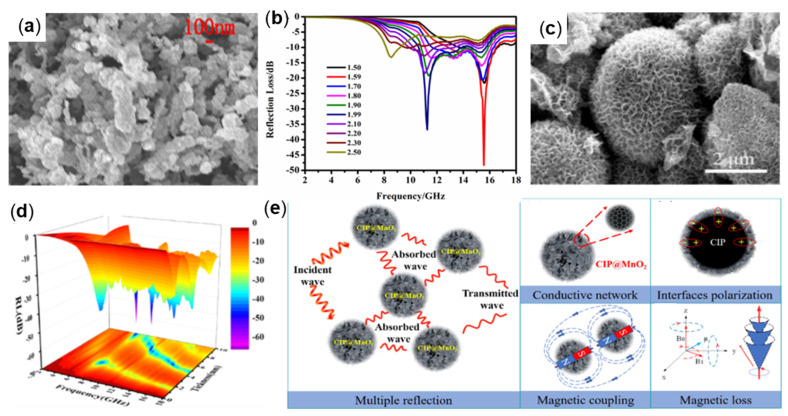
(**a**) SEM and (**b**) RL plot of Fe/ZnO, Reproduced with permission from Ref. [92], Copyright 2021, Elsevier. (**c**) SEM and (**d**) RL plot of CIP@MnO_2_ and (**e**) wave absorption mechanism of CIP@MnO_2_ composites with a spherical flower-like structure. Adapted with permission from Ref. [96].

**Figure 13 molecules-27-04117-f013:**
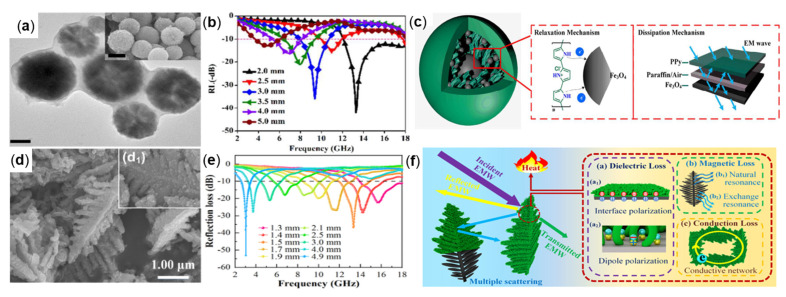
(**a**) SEM, (**b**) RL plot of Fe_3_O_4_@PPy, and (**c**) wave absorption mechanism of Fe_3_O_4_@PPy composites, reproduced with permission from Ref. [101], Copyright 2017, American Chemical Society. (**d**) SEM, (**e**) RL plot of Fe_3_O_4_@PANI, and (**f**) the EMW absorption mechanism of dendritic Fe_3_O_4_@PANI composites, reproduced with permission from Ref. [102], Copyright 2021, Elsevier.

**Table 1 molecules-27-04117-t001:** The electromagnetic absorption performance of sphere-like Fe.

Samples	Methods	ƒ_E_ (GHz)	Thickness (mm)	Filling Ratio	RL_min_ (dB)	Reference
Fe_3_O_4_	Pyrolysis	11.76	2.07	70 wt%	−55.14	[35]
PIPs	Corrosion technique	13.2	1.8	20 vol%	−42.2	[38]
Fe_4_N	Gaseous nitriding	3.5	3.0	75 wt%	−33	[40]

**Table 2 molecules-27-04117-t002:** The electromagnetic absorption performance of flaky-like Fe.

Samples	Methods	ƒ_E_ (GHz)	Thicknes (mm)	Filling Ratio	RL_min_ (dB)	Reference
FCI	Ball milling	2 6.2	-	70 vol%	*μ*′ = 1.61 *μ*″ = 3.20	[46]
Scale-like FCI	Ball milling	4.5~8.5	1.5	85 wt%	<−10 dB	[47]
PACI	Ball milling	2.09	3.25	70 wt%	−53.1 dB	[42]
FCI	Ball milling	8~18	1.47	50 wt%	<−10 dB	[48]
FCI	Ball milling	5.92~18	0.8	40 wt%	<−8 dB	[49]
FCIPs	Ball milling	2~18 26.5~40	6	75 wt%	<−10 dB	[50]

**Table 4 molecules-27-04117-t004:** The electromagnetic absorption performance of blended Fe-based composites.

Samples	Methods	ƒ_E_ (GHz)	Thickness (mm)	Filling Ratio	RL_min_ (dB)	Reference
CIPsPDMS	Doctor blade	14.6	1.5	72 wt%	−27.5	[62]
FeFA	Carbothermal reduction	16.1	2.5	65 wt%	−35.7	[63]
MWCNTs/Fe_3_O_4_ /PPY/C	Situ polymerization	13.92	2.2	25 wt%	−53.07	[64]
Fe_3_O_4_/C	Refluxing, annealing	12.2−17.8	1.9	25 wt%	−46.5	[65]

## Data Availability

Data of the compounds are available from the authors.

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
