# Peer review of "Recent Progress in Iron-Based Microwave Absorbing Composites: A Review and Prospective"

_molecules, 2022, doi:10.3390/molecules27134117_

Round 1

Reviewer 1 Report

Review of "Recent Progress in Iron-Based Microwave Absorbing

Composites"

The review discusses the idea of ​​preventing microwave pollution and the various materials and methods previously used for the same proposal.

The authors discussed the benefits of iron as an electromagnetic metal in absorbing electromagnetic waves and also showed the different morphological properties of iron metal and its ability to absorb radiation.

In this review, the authors explore the preparation methods, absorption properties, and absorption mechanism of iron-based adsorbents from two aspects: different forms of iron-based adsorbents and iron-based compound adsorbents.

The conclusion also summarizes the different modifications made with different authors in modifying iron metal morphology and the effect of different iron morphology on the absorption strength.

I see the review as an addition in the field of studying methods for protecting humans from microwave radiation and is also a good idea in military research. Review of "Recent Progress in Iron-Based Microwave Absorbing Composites" is a good collection and study in the field of microwaves absorption materials and the review is a good study introduced.

I agree with publishing of the review in its current form.

Author Response

Reply to reviewer #1 please see attachment

Reviewer 2 Report

What is the proposed physical mechanism for the effect of particle size on microwave absorption?

Author Response

Reply to reviewer #2 please see attachment

Reviewer 3 Report

Review of paper by Wei Zheng et al.

Recent Progress in Iron-Based Microwave Absorbing 2 Composites: A Review and Prospective

 This review paper is useful and well written and should be published subject to the additions of some data for the sake of completeness and accuracy because the authors severely restricted themselves in the scope of the subject and also somewhat ignored the recent developments.  The applications of electromagnetic wave absorption are extremely wide and it would not be in the interest of science and technology to threat each application in isolation.  This is my main concern and should be addressed before its acceptance.    

My comments are below.

 1.The acronym EMWAs does not match its narrative of the paper which defines EMWAs as  “electromagnetic wave absorbing materials”.  Hence the acronym  should be changed to EMWAMs or it should be re-defined.

2.Infact, there are so many acronyms used in the paper.  Some times one wonders if they refer to the correct text.  The paper will benefit to provide a list of acronyms.

3.The objective of the paper is described as: In this paper, the research progress of Fe-based absorbing materials is reviewed firstly, and then the preparation methods, absorbing properties, and absorbing mechanism of Fe-based absorbing materials are discussed in detail from two aspects: different morphologies of Fe-based absorbing agents and Fe-based composite absorbing agents. Finally, the future development direction of Fe-based absorbing materials is prospected.”

4.The  title of the paper indicates that the review is all about Fe-based microwave absorbing materials.  But in the introduction and subsequent parts, the scope of the review is broadened and the electromagnetic wave absorbing materials are considered with acronym EMWA  being used throughout the text.  It might be better to make connection between microwave and higher energy electromagnetic radiations in terms of frequencies.  This is all the more important as THz range frequencies are also used in some of the applications cited for microwaves, such as imaging.  Although the title is microwave absorption, in reality the review confines itself to radar frequencies. 

5.In search of efficient Fe-based absorbers, two methods are adopted for review: (1) Enhancement of the magnetic permeability and magnetic loss of the absorbers through structural design; (2) the use of multi-component material compounding.  However, this second approach does not clarify if this approach includes chemical multi-component materials (such as hexaferrites)  or is it confined to physical multi-compound materials (such as supported absorbers or shell/core structures). 

6.Looking at the subsequent text, it is clear that physical multi-compound materials are the focus of the review.   However, the disadvantage of this approach is that it removes some of the important recent progress in EM-radiation absorbing materials which can further benefit from the first approach in achieving even greater efficiency.

7.Furthermore, EM absorption, both in microwave range and at higher frequencies, are important in other applications, including chemical catalysis, imaging, detectors,  etc.  In these systems, the structural and chemical compositions of absorbers and catalysts are very similar.  Therefore, the review must cite several papers in these areas to indicate the existing of developments which are equally valid for microwave absorption technologies. 

8.This point is infact reflected in Section 2 of the review paper. Wave absorption mechanisms of EMWAs.    The review should include the following references:

a)    M. Green and X. Chen, Recent progress in nanomaterials for microwave absorption.  J. Materiomics, 5, 2019, 503-541.  (This is an excellent review of EM absorbing nanomaterials.)

 b)    B. Wang, Q. Wu, Y. Fu, T. Liu, A review on carbon/magnetic metal composites for microwave absorption, J. Mater. Sci. Technol., 86 2021, 91-109.  (This review is an update of the above publication focussing on carbon/graphene supports and metals, including Fe).  

 c)    R. C. Pullar, Hexagonal ferrites: A review of the synthesis, properties and applications of hexaferrite ceramics, Progress in materials Science, 57, 2012, 1191-1334.   (This excellent review covers an important class of Fe-based absorbers)

 d)    G. Akay, Plasma Generating—Chemical Looping Catalyst Synthesis by Microwave Plasma Shock for Nitrogen Fixation from Air and Hydrogen Production from Water for Agriculture and Energy Technologies in Global Warming Prevention. Catalysts, 10, 2020, 152.  (It introduces a generic multi-component, multi-structured catalysts and EM-absorbing materials which generates plasma under microwave radiation in air. Thus EM-absorption is through chemical reaction.)

 e)    G. Akay, Microwave generating-chemical looping catalyst synthesis.  World Intellectual Property Organisation Publication, WO2021/145843 A2, 2021.  (This patent contains the EM radiation absorption data for several types of absorbents (including Fe-based) produced under microwave induced plasma shock using precursor metal salts and dielectric and ferroelectric particles in water.  The resulting porous structures are highly heterogeneous with semiconductor and high metal/metal oxide regions at micrometre scale.  Their absorption coefficients of these materials are in the same order of magnitude as water. Variety of structures are observed including core/shell domains).

9.Although quantum effects are mentioned (with ref [29]) when the size of the absorption particles are in nano meter range, its manifestation and mechanism are not elaborated.

10.Figure 2 of the review manuscript, although it is a useful descriptive image, it is not complete.  Reference should be given to the paper/patent cited in Section 5(d,e) as a novel method of EM-radiation absorption by plasma generation and chemical reaction.  This mechanism of EM-wave absorption does not depend on frequency of radiation and the limited  data presented in 5(e) indicates a water-like behaviour in microwave absorption.  

11.The omission of hexagonal ferrites in the review, especially W-type barium hexaferrites reduces the impact of the review.

12.The paper will benefit from a list of acronyms as there are too many of them.    

Author Response

Reply to reviewer #3 please see attachment
